# Comparative Genomics Revealed a Potential Threat of *Aeromonas rivipollensis* G87 Strain and Its Antibiotic Resistance

**DOI:** 10.3390/antibiotics12010131

**Published:** 2023-01-09

**Authors:** Esther Ubani K. Fono-Tamo, Ilunga Kamika, John Barr Dewar, Kgaugelo Edward Lekota

**Affiliations:** 1Department of Life and Consumer Sciences, College of Agriculture and Environmental Sciences, University of South Africa, Florida Campus, Johannesburg 1709, South Africa; 2Institute for Nanotechnology and Water Sustainability (iNanoWS), School of Science, College of Science, Engineering and Technology (CSET), University of South Africa, Florida Campus, Johannesburg 1709, South Africa; 3Unit for Environmental Sciences and Management: Microbiology, North-West University, Potchefstroom Campus, Private Bag X6001, Potchefstroom 2520, South Africa

**Keywords:** *Aeromonas rivipollensis*, whole-genome sequencing, pangenomics, antibiotic resistance, mobile genetic elements

## Abstract

*Aeromonas rivipollensis* is an emerging pathogen linked to a broad range of infections in humans. Due to the inability to accurately differentiate *Aeromonas* species using conventional techniques, in-depth comparative genomics analysis is imperative to identify them. This study characterized 4 *A. rivipollensis* strains that were isolated from river water in Johannesburg, South Africa, by whole-genome sequencing (WGS). WGS was carried out, and taxonomic classification was employed to profile virulence and antibiotic resistance (AR). The AR profiles of the *A. rivipollensis* genomes consisted of betalactams and cephalosporin-resistance genes, while the tetracycline-resistance gene (*tetE*) was only determined to be in the G87 strain. A mobile genetic element (MGE), transposons *TnC*, was determined to be in this strain that mediates tetracycline resistance MFS efflux *tetE*. A pangenomic investigation revealed the G87 strain’s unique characteristic, which included immunoglobulin A-binding proteins, extracellular polysialic acid, and exogenous sialic acid as virulence factors. The identified polysialic acid and sialic acid genes can be associated with antiphagocytic and antibactericidal properties, respectively. MGEs such as transposases introduce virulence and AR genes in the *A. rivipollensis* G87 genome. This study showed that *A. rivipollensis* is generally resistant to a class of beta-lactams and cephalosporins. MGEs pose a challenge in some of the *Aeromonas* species strains and are subjected to antibiotics resistance and the acquisition of virulence genes in the ecosystem.

## 1. Introduction

*Aeromonas* species are considered autochthonous to aquatic environments and cause a range of opportunistic infections in humans [1]. They are emerging, opportunistic pathogens that frequently transmit from the environment to humans, causing a wide spectrum of infections [2]. They are characterized as ubiquitous Gram-negative bacilli that consist of a genetic population structure with several characteristics that favor an evolutionary mode of species complexes that are heterogeneous [3]. This group comprises closely related species of distinct strains that are influenced by horizontal gene transfer (HGT) [4].

In previous studies, *Aeromonas* species have been detected in marine, estuarine, and freshwater systems [1,5]. They have also been identified as etiological agents of infections in aquatic animals and humans [6] and are often associated with gastroenteritis and wound infections in humans [7,8]. Several reports linked *Aeromonas* spp. to fish and cold-blooded aquatic animal infections, such as septicaemia, keratitis, open wounds, and ulcers [2,9]. However, in recent times, new potentially pathogenic species such as *A. rivipollensis* have been detected [10]. Marti and Balcázar [11] described *A. rivipollensis* for the first time and classified it as closely related to *A. media* based on the multilocus sequence typing of five housekeeping genes (*gyrA*, *gyrB*, *recA*, *dnaJ*, and *rpoD*). However, it is closely related to both *A. media* and *A. hydrophila* when 16S rRNA phylogenetic analysis is employed. Nevertheless, its genotypic characterization is still incomplete and not fully exploited. A complete genome of *A. rivipollensis* is well described and suggests a zoonotic potential like that of other aeromonads [10].

In most bacterial species, insertion sequences (IS) and transposons (Tn) are examples of mobile genomic elements (MGEs) that promote the spread of antibiotic and virulence genes [12]. The mediation of MGEs often leads to bacterial evolution and also alters phenotypes [13]. This poses a threat, as *Aeromonas* species show a complex heterogeneous strains because they show various metabolic capabilities to adapt to their environmental change, which results in the acquisition of numerous virulence factors [4,14,15]. The *Aeromonas* species comprises a wide range of virulence factors that are involved in biofilm formation, invasion, cell adherence, and cytotoxicity in polar and lateral flagella [16,17], lipopolysaccharides [18], adhesins [19] iron-binding systems [20], and other extracellular toxins and enzymes [21,22].

Although *A. rivipollensis* is considered an emerging species, it is possible that the inability of current conventional identification methods to efficiently differentiate *Aeromonas* species might have masked the reported prevalence as well as the characterization of this organism [5]. The use of mass spectrometry–time of flight (MALDI–TOF MS) and 16S rRNA gene sequencing has proven to be inadequate in differentiating some of the *Aeromonas* species [23]. The reliability of using whole-genome sequencing fully examines the antibiotic resistance and virulence genes, especially in this heterologous species of *Aeromonas* spp. The evolution of this genus also shows limited information about the phylogenomic structure in *A. rivipollensis*, as few genomes are available. The genetic structure of *A. rivipollensis* is unknown, and this species appears to be a heterogeneous phylogenetic cluster that could pose a threat to humans. Thus, studying its genomic features is imperative to understanding its antibiotic resistance and virulence genes. This study sequenced four *A. rivipollensis* genomes to determine the virulence and antibiotic-resistance genes, as well as to define its genetic population structure.

## 2. Results

### 2.1. Genome Features of Aeromonas rivipollensis

Prior to genomic assessment, isolated strains were identified as *Aeromonas* spp. based on their morphological characteristics on the selective *Aeromonas* isolation agar. The MALDI–TOF method was used to further identify the isolates designated as G36, G42, G78, and G87 strains, which were then determined to be closely related to *A. media*. However, when whole-genome sequencing was employed, the isolates were identified as *A. rivipollensis*. The assembled genomes’ quality assessment revealed these isolates had an estimated completeness of more than 99%. The genome sizes (Mb) of the G36, G42, and G78 strains were approximately 4.53, 4.58, and 4.53 Mb, respectively (Table 1). The genome size of the *A. rivipollensis* G87 strain is 4.66 Mb, which is slightly higher than the compared genomes in this study. The protein coding sequences (CDS) of the G36, G42, G78, and G87 genomes consist of 4,239, 4,273, 4,205, and 4,319, respectively. The G + C content of the sequenced *A. rivipollensis* genomes is 61%, identical to *A. rivipollensis* KN-Mc-11N1. The high number of CDSs in *A. rivipollensis* G87 included transposon *Tn7* transposition proteins (*TnsA*, *TnsB*, and *TnsC*), ribosomal protein S12 methylthiotransferase (*RimO*), galactokinase (*GalK*), plastocyanin (*PetE*), copper-resistance protein *CopA*, and cobalt–zinc–cadmium resistance protein *CzcA*. The high number of tRNAs (*n* = 98) was also observed in the *A. rivipollensis* G87 genome.

### 2.2. Taxonomic Classification Using gyrB and Whole-Genome-Based Species Tree

The *gyrB* marker was used to identify the *Aeromonas* spp. isolates investigated in this study (Appendix A). The isolates were identified as *A. rivipollensis* (Appendix A) and grouped with *A. rivipollensis* strain KN-Mc-11N1. The 94-bootstrap value of *A. rivipollensis* distinguishes it from the other *Aeromonas* species. The use of 16S rRNA gene sequencing did not significantly classify the *A. rivipollensis* strains. This species is heterologous since the genus *Aeromonas* contains many phylogroups. Using the whole-genome species tree (Figure 1), it was shown that *Aeromonas* spp. clustered apart from the Enterobactericeae family, which includes *E. coli*, *Salmonella* serovars, and other species. The isolates grouped with previously sequenced genome *A. rivipollensis* KN-Mc-11N1. This demonstrates that the genomes of *A. rivipollensis*, *A. eucroniphilla* strain CECT 4224, *A. salmonicida* subsp. *salmonicida* strain A449, and *A. mulluscorum* strain 848 share some of the core genes.

### 2.3. Pangenome Analysis of the Aeromonas Species

A total of 23,119 genes were examined and defined by the pangenome analysis of the 15 *Aeromonas* spp., with 375 core genes (found in >99% of genomes) shared by the *Aeromonas* species of *rivipollensis*, *salmonicida*, *molluscorum*, and *euroniphilla* (Figure 2A). Most genes were identified as accessories and consisted of 8899 and 13,845 shell and cloud genes, respectively. This indicates that *Aeromonas* is a heterogeneous species that consists of many cloud genes (Appendix A). *A. rivipollensis* strains clustered significantly in their own sub-clade (Figure 2 and Appendix A). The different strains of *A. rivipollensis* are shown by their distinctive average nucleotide identity (ANI) profiles identified using accessory genes (Figure 2B). This observation is also an augment as determined by whole-genome species tree. *Aeromonas* species genomes contained core metabolic enzymes, such as amidophosphoribosyltransferase (*purF*), ATP-dependent 6-phosphofructokinase isozyme 1 (*pfka*), nitrogen regulatory protein (*ptsN*), glutamate-pyruvate aminotransferase (*alaA*), UDP-3-O-acyl-N-acetylglucosamine deacetylase (*ipxC*), peptidyl-prolyl cis-trans isomerase B (*ppiB*), maltose transport system permease (*maIG*), cysteine desulfurase (*iscS*), dihydrolipoyl dehydrogenase (*ipdA*), UDP-N-acetylglucosamine 1-carboxyvinyltransferase (*murA*), and oligopeptide transport system permease (*oppC*). The accessory binary genes (Figure 2B) found in *A. rivipollensis* strain G87 were biosynthetic genes for polysialic acid, which is responsible for capsulation and is also found in the KN-Mc-11N1 strain.

The accessory metabolic genes on *A. rivipollensis* strain G87 included genes such as taurine import ATP-binding protein (*tauB*) and taurine-binding periplasmic proteins, cellulose synthase operon protein C (*bcsC*), beta-xylosidase (*xynB*), glucose-6-phosphate isomerase (*pgiA*), unsaturated chondroitin disaccharide hydrolase (*ugi*), p-aminobenzoyl-glutamate transport protein (*abgT*), morphine 6-dehydrogenase (*morA*), beta-glucoside kinase (*bglK*), and aryl-phospho-beta-D-glucosidase (*bglC1* and *bglC2*).

### 2.4. Polysialic Acid and Sialic Acid Biosynthesis Genes

In the *A. rivipollensis* G87 and KN-Mc11N1 phylogroup, we have identified a polysialic acid operon that consists of kpsM, kpsT, kpsE, kpsD, and KpsFS (Table 2). These are capsular polysaccharide genes that are associated with polysialic acid in the outer membrane. The gene cluster organisation (Figure 3) is also outlined, which shows different kps genes and precursor components (neuABC) for the biosynthesis of extracellular polysialic acid. The kps cluster comprises two different regions, namely: region 1 (kpsDMTE) and region 3 (neuC and kpsFS) (Figure 3). The extracellular kpsDMTE genes and kpsFS are required for the polysaccharide capsule formation in the bacteria (blue and red). The kpsFS genes participate in the translocation of the polysaccharide capsule. Pangenome analysis showed that these genes are absent in the other sequenced *A. rivipollensis* in this study. Genome strain G87 can be separated from the KN-Mc-11N1 strain based on genes responsible for exogenous sialic acid production.

*A. rivipollensis* strain G87 was the only genome with an exogenous sialic acid metabolism. This was further confirmed by the RAST annotation server, which assigned the subsystem a score of 2.0. The metabolic cluster for sialic acid is composed of different nan-genes (Figure 3). The annotation names for the polysialic acid biosynthesis cluster (kps) and sialic acid metabolic cluster were assigned to the *A. rivipollensis* strain G87 (Table 2). The nanARH and TRAP transporters (nanTp-TI) (green) are responsible for exogenous sialic acid, while the nanAXEKP and yhcH genes (orange) are responsible for the degradation or catabolism of salic acid.

### 2.5. Antibiotics and Virulence Genes of Aeromonas rivipollensis Genomes

In this study, several core-virulence genes (*n* = 7) from the sequenced *Aeromonas rivipollensis* were identified (Figure 4). The chemoreceptor tsr gene, flagellar biosynthesis proteins *fliACN*, and twitching motility proteins *pilBJT* were among them. The antibiotic resistance genes shared by the *A. rivipollensis* genomes included cephalosporins (*ampH*) and beta-lactams (*blaCMY*) genes. Tetracycline resistance (*tetE*) gene was the unique antibiotic-resistance gene present in the *A. rivipollensis* G87 genome.

### 2.6. Mobile Genetic Elements

In this G87 strain containing the transposition protein TnsC, a tetracycline-resistance MFS efflux pump known as tetE and a tetracycline-resistance regulatory protein tetR were discovered (Appendix A). The transposition protein TnsC had a BLAST nucleotide homology of 98.70% with *A. veronii* strain wp8-s18-ESNL-11. The hypothetical proteins identified in this gene cluster also had a 100% nucleotide identity with *A. veronii* strain wp8-s18-ESNL-11. In the upstream region in this operon, a Type I restriction–modification system specific subunit S and DNA–methyltransferase subunit-M were determined to influence genomic evolution by horizontal gene transfer. A transposase, InsH for insertion sequence element IS5, was also determined to be in *A. rivipollensis* strain G87. In the downstream part of the gene arrangement cluster, this InsH transposase also features a secretory immunoglobulin A-binding protein (esiB1) of 726 bp (242 aa). The second esiB2 gene of about 1656 bp (522 aa) was also found lying downstream with other mobile genetic elements. The IS3 family transposase ISAs7 (618 bp) was also found that had a BLASTn of 98.54% and 98.22% with A. media strain T0.1-19 and *A. rivipollensis* strain KN-Mc-11N1, respectively. Other MGEs identified were two transposon Tn3 resolvase proteins (tnpR1 and tnpR2) and Tn7 found in the G87 strain.

## 3. Materials and Methods

### 3.1. Sample Isolation and Classical Microbiological Tests

We collected water samples from the Jukskei River continuum in Johannesburg (25.948156 S, 27.957528 E) in 2018. Water samples were aseptically collected in duplicates of approximately 1 L in a sterile container. Water samples were immediately placed in a cooler box at 4 °C and transported to the laboratory for further analysis. For bacterial isolation, 100 mL of the sample was filtered through a nitrocellulose filter membrane (0.22 μm), and the filters were then placed on *Aeromonas* isolation agar (Merck, Millipore) and incubated at 37 °C for 24 h. Four distinct isolates were coded as G36, G42, G78, and G87 and were presumptive *Aeromonas* species. The presumptive isolates were further identified using matrix-assisted laser desorption ionization–time-of-flight mass spectrometry (MALDI–TOF) (Bruker, Bremen, Germany). Briefly, a single colony was transferred into an Eppendorf tube containing 300 μL of sterile distilled water and 900 μL of absolute ethanol, mixed thoroughly, and later centrifuged for 2 min at 13,000 rpm. The supernatant was discarded, and the Eppendorf tube was later filled with 5 μL of 70% formic acid and thoroughly mixed.

### 3.2. DNA Extractions

The genomic DNA of the *Aeromonas* species strains was extracted using the High Pure PC Template preparation kit (Roche, Germany). The DNA was quantified on qubit fluorometric quantization using the Broad Range assay kit (Invitrogen™). The quality of the DNA was analyzed by electrophoresis on a 0.8% agarose gel using ethidium bromide and visualized under UV-light.

### 3.3. Whole-Genome Sequencing, Quality Control and Assembly

The sequencing libraries of the *Aeromonas* isolates (*n* = 4) were generated using the NEBNext^®^ Ultra^TM^ II FS DNA library prep kit (New England Biolabs). The 150 bp paired-end sequence reads were generated with the Illumina MiSeq sequencer (Illumina, USA). FastQC v. 0.11.52 (http://www.bioinformatics.babraham.ac.uk/projects/fastqc (accessed on 6 February 2022)) was used to assess the quality of the paired-end reads associated with each isolate. Sequence coverage of the genomes of *Aeromonas* isolates G36, G42, G78, and G87 was 39X, 144X, 54, and 294X, respectively. Trimmomatic v. 0.33 [24] was used to remove the sequenced adapters and the leading and trailing ambiguous bases. The trimming parameters included LEADING:3 and TRAILING:3; reads with average per-base quality scores of <15 within a 4 bp sliding window. The SLIDINGWINDOW:4:15; reads with length < 36 bp were removed. Trimmed reads were de novo assembled using the CLC Genomics Workbench v. 11.01 (Qiagen). QUAST v. 4.5 [25] was used to assess the quality of the associated assembled genome using default parameters. CheckM [26] was additionally used to assess the potential contaminants in each assembled genome of *Aeromonas* isolates using default parameters. BLASTn [27] was used to align the assembled contigs using *A. rivipollensis* KN-Mc-11N1 [10] as a reference. For consistency and the removal of contaminants, multiple genome alignments were constructed using the progressive MAUVE tool [28]. The feature prediction and annotation of the sequenced genomes were performed using the NCBI prokaryotic Genome Annotation pipeline (PGAP) [29] and rapid annotation RAST [30].

### 3.4. Phylogenetic Analysis Using gyraseB and Whole-Genome Species Tree

The gyraseB (*gryB*) gene sequences were extracted from the assembled genomes of the *A. rivipollensis* strains. The nucleotide sequence queries of the *Aeromonas* isolates were BLAST-searched and compared with the available sequences of *Aeromonas* species in NCBI (http://www.ncbi.nlm.nih.gov (accessed on 25 April 2022)). This enabled the evaluation of homologous hits to the NCBI’s available sequences. Multiple sequence alignments of the extracted gene sequence and mined NCBI sequences were performed using MAFFT [31]. Maximum likelihood phylogenetic analysis of *A. rivipollensis* was performed using 1000 bootstrap iterations in MEGA 11.0 [32]. Utilizing K-base’s species tree (https://www.kbase.us/ (accessed on 12 March 2022)), the in silico taxonomic classification of all four sequenced genomes was performed. The assembled genomes were compared to other genomes that were accessible in the NCBI database. The whole-genome phylogenetic tree was visualized using Figtree v1.16.6 (http://tree.bio.ed.ac.uk/software/figtree/ (accessed on 25 March 2022)).

### 3.5. Pangenomics Analysis

The inclusion of the *Aeromonas* spp. genomes (*n* = 11) was based on the species tree generated from whole-genome-based taxonomic classification. This study included the *A. veronii* and *A. hydrophilla* genomes based on their completeness status. *A. veronni* genomes included the South African strains isolated from human cases, and none were found from the environment. In total, about 15 *Aeromonas* spp. genomes were used for comparative genomic analysis that includes the 4 sequenced *A. rivipollensis*. Prokka v.1.14.0 [33] was used to annotate the genomes of *Aeromonas* species using the default parameters. The pan-genome composition was extracted using Roary [34]. Pan-genome clusters were defined as follows: Core-genes were present in all isolates; soft core-genes present in at least 95% of isolates; shell-genes were present between 15% and 95% of isolates; cloud-genes were present in less than 15% of isolates. IQ-TREE v.2 [35] was used to construct the phylogenetic tree of the aligned core-genes using default parameters. The core genome phylogenetic tree was visualized using the Newick tree display [36]. Additionally, phandango v1.3.0 (https://jameshadfield.github.io/phandango/#/ (accessed on 25 April 2022)) was used for the interactive visualization of the genome phylogeny.

### 3.6. Antibiotics Resistance, Plasmid Replicon, Mobile Genetic Elements, and Virulence Factor Determinants

Antibiotic-resistance genes were identified in four sequenced *Aeromonas* genome sequences and the reference strain. In the ABRicate pipeline, AMR determinants were identified in each assembled genome using the ResFinder database (–db ResFinder; accessed 23 April 2022) [37], which has minimum identity and coverage thresholds of 75 (–minid 75) and 50% (–mincov 50), respectively. Plasmid replicons were identified on the sequenced genomes using the PlasmidFinder database [38]. Resistance genes were determined using the Virulence Factor Database (VFDB; –db vfdb, accessed 19 April 2022) [39], using minimum identity and coverage thresholds of 70 (–minid 70) and 50% (–mincov 50), respectively. OriTfinder v1.1 [40] was also used to predict the virulence factors and acquired antibiotic resistance genes. Mobile genetic elements (MGEs) were investigated on the sequenced genomes of *Aeromonas* spp. using this tool as well as default parameters. The heatmap of the antibiotics resistance and virulence genes was generated using GraphPad Prism9.

## 4. Discussion

The identification of *Aeromonas* spp. isolates to the species level remains a challenge using classic microbiological tests. Without a suitably comprehensive database, methods that are most frequently used, like MALDI–TOF, may misidentify samples [23]. However, this study used a high-resolution whole-genome and pangenomics analysis to confirm and identify *A. rivipollensis* genomes. This study showed that *A. rivipollensis* seems to be an emerging pathogen that has not yet been thoroughly characterized. The observation of accessory genes that included polysialic acid and sialic acid, mobile genetic elements, and an antibiotic-resistance profile, especially in the genome of G87, elucidate a unique genetic cluster of *A. rivipollensis*.

The use of *gyrB* was used in this study to fairly classify and identify *A. rivipollensis*. The sequenced *Aeromonas* species could only be correctly identified as *A. rivipollensis* using the high-resolution method of WGS and whole-genome species tree. These isolates were identified as *A. media* using MALDI-TOF. They were grouped with *A. rivipollensis* KN-Mc-11N1 using the WGS species tree (Figure 1), and pangenome analysis. Their genomes are quite closely related to those of *A. eucroniphilla* strain CECT 4224, *A. salmonicida* subsp. *salmonicida* strain A449, and *A. mulluscorum* strain 848. The pangenomic examination of this study’s species revealed that the genus *Aeromonas* is diverse when compared to other species, since only a small number (*n* = 375) of core genes could be identified. Distinct clusters are also observed among *A. rivopollensis* genomes. The genomes of KN-Mc-11N1 and G87, which share several core genes, make up one of the clusters. This cluster contained the polysialic acid (PGA) (*kpsDMTE*) genes responsible for exogenous capsulation, whereas sialic acid was only found in G87, which is associated with virulence and adaptation in the environment [41].

*The primary strains for polysialic acid (PSA) biosynthesis and metabolism were A. rivipollensis G87 and KN-Mc-11N1*. Many other bacteria, such as *Mannheimia hemolytica* (previously *Pasteurella haemolytica*) [42] and *E. coli* [43,44], have been associated with the production of PSA as their extracellular capsules. In bacteria, the PSA acts as a virulence factor that mimics the mammalian PSA’s structure and as an antiphagocytic [45]. The *kps* gene cluster that is involved in the PSA biosynthesis, modification, and transport of the bacteria’s PSA chains is outlined in this study for *A. rivipollensis* (Figure 3). The *kps* cluster comprises of two different regions, namely; region 1 (*kpsDMTE*) and region 3 (*neuC* and *kpsFS*), which participate in the translocation of the polysaccharide across the periplasmic space and onto the cell surface [44,46]. PSA metabolism is regulated at a transcriptional level by the transcriptional activator *rfaH* [47]. In *E. coli* K92, *rfaH* enhanced *kps* expression for the synthesis of the polysialic acid capsule [43]. This gene was also present in the genome of *A. rivipollensis* G87 at the downstream end of the operon. The polysialic acid capsule is transported across the outer membrane to the cell surface by the KpsD protein. It functions as the periplasmic binding element of the PSA transport system, in which it transiently interacts with the membrane component of the transporter, binds polysaccharide, and transports the polymer to a component in the outer membrane. This is also observed in bacteria containing poly-gamma-glutamate transpeptidase, responsible for capsulation [48]. Other components reported included the *kpsT* and *kpsM* genes, which are ABC transporters that export PSA from the cytoplasm. These transporters also require *kpsE*, a polysaccharide export system located in the inner membrane protein [46]. However, the mechanism of expressing the polysialic is not well understood and characterized in *A. rivipollensis*, as this is the first report in this phylogroup of G87 and KN-Mc-11N1 genomes that needs further investigation.

A complete nan system was outlined as including at least one ortholog of each of the genes encoding *nanA*, *nanE*, and *nanK*, more especially in the *E. coli* model [49]. These genes were absent from the other sequenced and compared genomes in this study but were present in the *A. rivipollensis* strain G87 genome. This was discovered in the annotation subsystem and was confirmed as an exogenous sialic acid based on the genetic variant code 2.0 assigned to it. Sialic acid is biosynthesized, activated, and polymerized by proteins NeuABCD [50], which are present in the G87 genome with the exception of the *neuD* gene. The *neuD* is a gene found in organisms that can synthesize sialic acid [51].

It is well recognized that the precursor to PSA is the gene for N-acetylneuraminate (Neu5Ac aldolase or *neuC* gene). By mimicking sialic acid, some pathogenic bacteria can circumvent host defenses. The *neuC* is located upstream with other *kpsFS* genes, which is involved in the production of capsular heptose. Exogenous sialic acid nan operon *nanRAHTpTI* (region 4) and degradation nan operon *nanAXEKP-YhcH* (region 5) were assigned in Figure 3. De novo synthetic genes of the sialic acid *neuABC* (region 2/3) were identified in this operon. However, according to the annotation genetic variation code, the G87 strain displayed a non-synthetic sialic acid. Therefore, organisms that can catabolize but not synthesize salic acids are classified as exogenous sialic acids. The sialic acid operon contained the two tripartite ATP-independent periplasmic (TRAP) transporters, *nanTp and nanTI*, respectively. The *nanTp* is a major permease component of the TRAP-type C4-dicarboxylate transport system (Figure 3). The TRAPs are well studied as the key transporters involved in the uptake of sialic acid and are associated with roles in pathogenicity [52,53]. A second N-acetylneuraminate lyase (*nanA2*) was identified in the upstream with transcriptional regulator *nanR*, which controls the expression of proteins involved in sialic acid absorption and metabolism [54,55]. Other salic acid genes that were not well characterized in this study might also be suggested by the region 2 putative proteins. It has now been established beyond doubt that microbial sialic acid metabolism contributes to the pathogenicity of a variety of infectious illnesses. Many vertebrate cells’ surfaces include glycan molecules with sialic acid occupying the terminal position. Sialic acid engages in a number of biological processes, such as cell signaling and intercellular adhesion [56]. Pathogenic bacteria that coat themselves with sialic acid, such as Group B *Streptococcus*, offer resistance to components of the host’s innate immune response. These bacteria have evolved to utilize this substance to their advantage [41,57]. Two secretory immunoglobulin A-binding proteins EsiB (esiB1) and EsiB2 found in the G87 genome were among the other virulence factors discovered in this study. They have different sequence lengths, with *esiB1* being 726 bp (242 aa) and *esiB2* being 1656 bp (522 aa), respectively. Pathogenic strains are primarily composed of the *esiB* genes, which are known to impair neutrophil activation [58].

*Aeromonas* species are generally resistant to betalactams (*blaCMY*), and this was also observed in this study. The presence of *blaCMY* and *ampH* genes in all isolated *A. rivipollensis* was not a surprise, as they have been recorded in several bacteria isolates including *Aeromonas* species [59,60]. In addition, the genome of isolate G87 was only one of the sequenced *A. rivipollensis* in this study that showed a multidrug tetracycline-resistance gene. This AR gene was found in this genome because it was linked to a mobile genetic element, *TnC*, with a nucleotide percentage identity similar to that of *A. veronii*. However, tetracycline resistance has previously been reported in *A. veronii* strain MS-17-88 recovered from channel catfish [12,61]. It has also been reported from *Aeromonas* species isolated from South African aquatic environments [62].

Flagellar biosynthesis proteins FliACN and twitching motility proteins PilBJT (Type IV pilus) were found in all *A. rivipollensis* strains sequenced in our study. These genes are commonly found in *Aeromonas* species that are associated with pathogenicity for colonization [63]. The chemoreceptor *tsr* gene was also present in the sequenced *A. rivipollensis* strains. In the study, Oh et al. [64] also detected this gene in *E. coli*. This has also been confirmed in other studies, indicating that *E. coli* and *S. enterica* serovar *Typhimurium* genomes possess many chemoreceptor genes, *tsr* [65]. Many chemoreceptor genes have also been reported in many other microorganisms [65,66]. However, no study has reported the presence of the *tsr* gene in *A. rivipollensis*.

## 5. Conclusions

Using WGS analysis, we were able to determine the resistome of *Aeromonas rivipollensis*. All *A. rivipollensis* strains show potential resistance to various antibiotic lactamase classes, indicating that these aeromonads are potentially virulent emerging pathogens that can be transmitted by river water. The determined polysialic acid and sialic acid genes in the G87 genome can be associated with antiphagocytic and anti-bactericidal properties, respectively. Mobile genetic elements, such as transposases, introduce virulence and resistance genes, such as secretory immunoglobulin A-binding proteins and multidrug tetracycline genes in the *A. rivipollensis* G87 genome. These genomes will be used as a resource for additional research that may reveal new information on the genes responsible for accurate identification, pathogenic potential, and the relative health risks posed by environmental strains of *A. rivipollensis*.

## Figures and Tables

**Figure 1 antibiotics-12-00131-f001:**
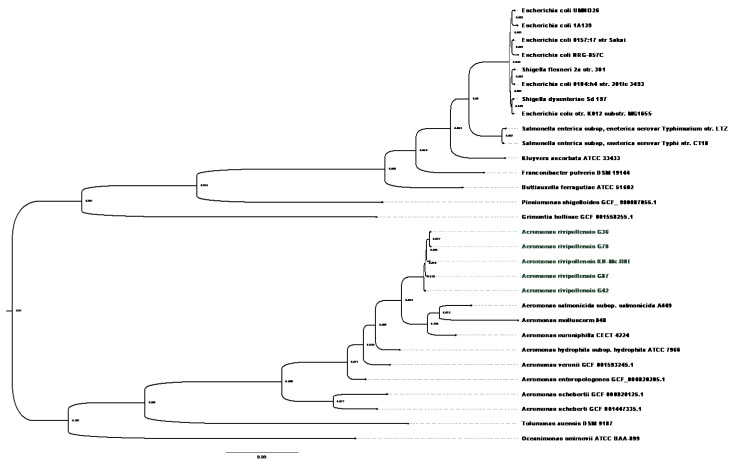
Whole-genome-based species tree showing the clustering of the sequenced *Aeromonas rivipollensis* genomes and KN-Mc-11N1 strain (green annotation) with the closely related genome species of *A. salmonicida* subsp. *salmonicida* A449, *A. molluscorum* 949, and *A. euroniphilla* CECT 4224.

**Figure 2 antibiotics-12-00131-f002:**
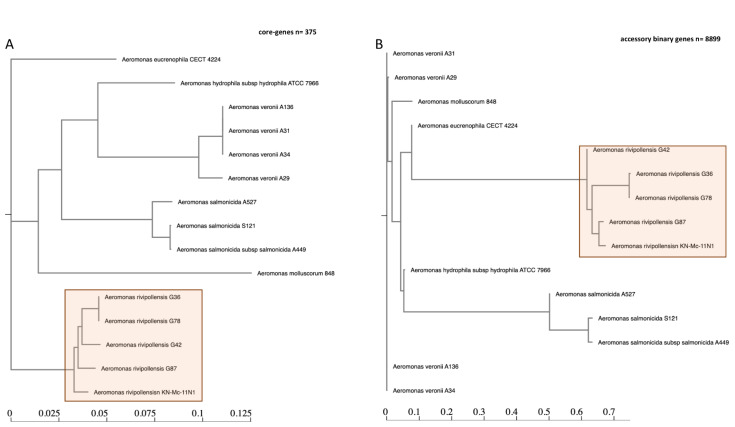
Phylogeny of the compared *Aeromonas* species using 375 core genes (**A**) and 8899 accessory binary genes (**B**). The tree shows phylogenetic tree of the 4 *Aeromonas* species as highlighted with orange color. About 375 core-genes (**A**) and 8899 accessory binary genes (**B**) were used to construct the phylogenetic tree.

**Figure 3 antibiotics-12-00131-f003:**
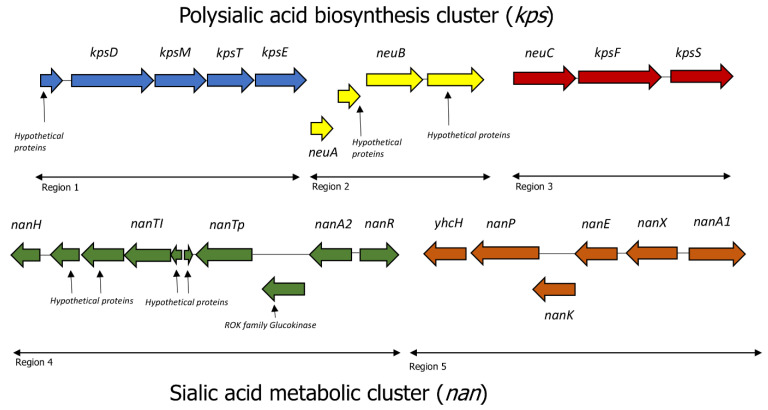
*Aeromonas rivipollensis* strain G87 organization gene clusters for capsular polysialic acids and an exogeneous sialic acid metabolic cluster. The *kpsDMTE* and *kpsFS* genes are required for capsulation (region 1, blue, and region 3, red). The *kpsFS* genes participate in the translocation of the polysaccharide. The precursors of polysialic acid biosynthesis are *neuABC* (region 2, yellow, and region 3, red). The *nanARH* with TRAP transporters in region 4, and *nanTp-TI* genes (green) are responsible for exogenous sialic acid, while the *nanAXEKP* and *yhcH* genes (orange) are responsible for the degradation or catabolism of salic acid.

**Figure 4 antibiotics-12-00131-f004:**
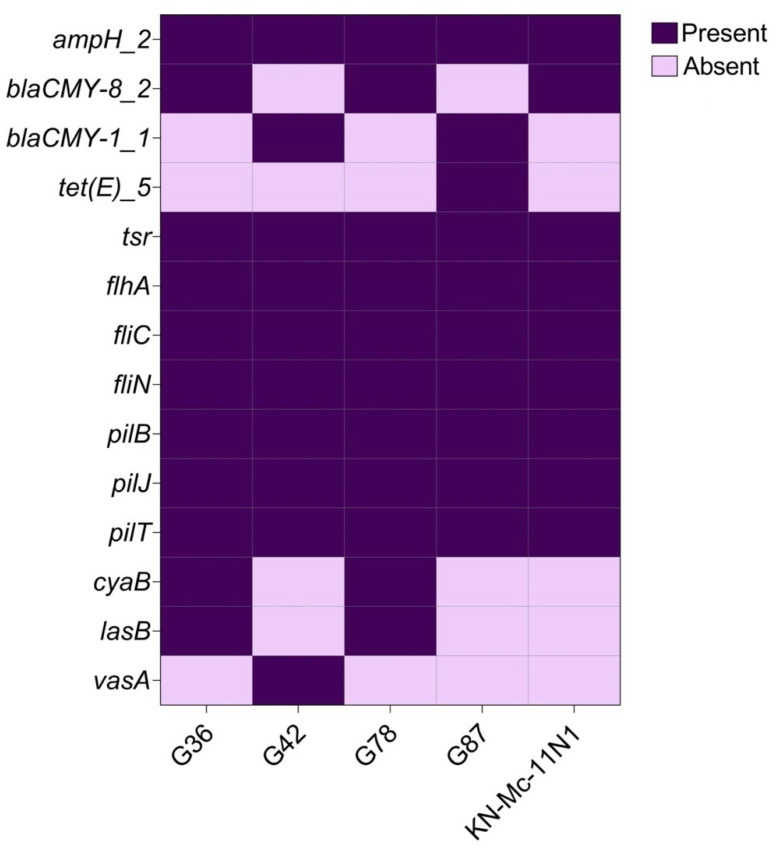
Heat map showing the antibiotic resistance and virulence genes detected among the 4 sequenced and the compared *A. rivipollensis* strain KN-Mc-11N1.

**Table 1 antibiotics-12-00131-t001:** Genome features of the sequenced *A. rivipollensis* strains used in this study compared to KN-Mc-11N1 as a reference strain.

Feature	G36	G42	G78	G87	KN-Mc-11N1 *
Genome size (bp)	4,530,639	4,584,495	4,531,506	4,663,030	4,508,901
CDSs (protein coding sequences)	4,239	4,273	4,205	4,319	4,025
Number of tRNAs	115	109	101	108	124
Number of total rRNAs	5	3	7	4	31
GC%	61,48	61,3	61,48	61,08	61,9
Number of contigs	93	55	57	71	1
N50	94,736	184,598	194,165	160,487	-
Sequence reads archive (SRA) or GenBank	SRR13249124	SRR13249123	SRR13249122	SRR13249121	CP027856.1

* Complete genome of *A. rivipollensis*—Not applicable.

**Table 2 antibiotics-12-00131-t002:** Gene annotation of the polysaccharide (polysialic acid) capsular and sialic acid genes involved in biosynthesis present in the genome of *A. rivipollensis* strain G87.

Gene Annotation	Abbreviation	Subsystem Assigned
Capsular polysaccharide export system periplasmic protein	*kpsD*	Capsular polysaccharide (CPS) of *Campylobacter* CPS biosynthesis and assembly
Capsular polysaccharide ABC transporter, permease protein	*kpsM*	Rhamnose containing glycans, CPS of Campylobacter, CPS biosynthesis and assembly
Capsular polysaccharide ABC transporter, ATP-binding protein	*kpsT*	CPS biosynthesis and assembly
Capsular polysaccharide export system inner membrane protein	*kpsE*	CPS of *Campylobacter*; CPS biosynthesis and assembly
COG3563: Capsule polysaccharide export protein	*KpsF*	CPS biosynthesis and assembly
Capsular polysaccharide export system protein	*kpsS*	CPS biosynthesis and assembly
N-Acetylneuraminate cytidylyltransferase (EC 2.7.7.43)	*neuA*	CMP-N-acetylneuraminate_biosynthesis; Sialic acid metabolism
N-acetylneuraminate synthase (EC 2.5.1.56)	*neuB*	CMP-N-acetylneuraminate biosynthesis; Sialic acid metabolism
dTDP-4-dehydrorhamnose 3,5-epimerase (EC 5.1.3.13)	*neuC*	dTDP-rhamnose_synthesis; Rhamnose containing glycans; Capsular_heptose_biosynthesis
Transcriptional regulator NanR	*nanR*	Sialic acid metabolism
N-acetylneuraminate lyase (EC 4.1.3.3)	*nanA2*	Sialic acid metabolism
TRAP-type C4-dicarboxylate transport system, periplasmic component	*nanTp* or *siaM*	TRAP Transporter collection
TRAP-type C4-dicarboxylate transport system, large permease component	*nanTI*	TRAP Transporter collection
Sialidase (EC 3.2.1.18)	*nanH*	Galactosylceramide and sulfatide metabolism; Sialic acid metabolism
N-acetylneuraminate lyase (EC 4.1.3.3)	*nanA1*	Sialic acid metabolism
Sialic acid utilization regulator, RpiR family	*nanX*	Sialic acid metabolism
N-acetylmannosamine-6-phosphate 2-epimerase (EC 5.1.3.9)	*nanE*	Sialic acid metabolism
N-acetylmannosamine kinase (EC 2.7.1.60)	*nanK*	Sialic acid metabolism
Predicted sialic acid transporter	*nanP*	Sialic acid metabolism
Putative sugar isomerase involved in processing of exogenous sialic acid	*yhcH*	Sialic acid metabolism

## Data Availability

*A. rivipollensis* G36, G42, G78, and G87 genome sequences were deposited in NCBI GenBank under the accession numbers JAAILC0000000001, GCA_010974915.1, GCA_010974825.1, and JAAIKZ000000000, respectively. The sequence read achieve (SRA) accession numbers are assigned as G87 (SRR13249121), G78 (SRR13249122), G42 (SRR13249123), or (G36 (SRR13249124).

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
