# Peer review of "Comparative Genomics Revealed a Potential Threat of Aeromonas rivipollensis G87 Strain and Its Antibiotic Resistance"

_antibiotics, 2023, doi:10.3390/antibiotics12010131_

Round 1

Reviewer 1 Report

Review of 'Comparative genomics revealed a potential threat of Aeromonas rivipollensis G87 strain and its antibiotic resistance' by Fono-Tamo et al.

The authors have carried out genomic sequencing on a collection of Aeromonas isolates and have identified virulence and antibiotic resistance genes in these genomes.

The questions being addressed by the authors on (i) the species identification of the isolates and (ii) the relatedness of the sequenced strains to one another and to previously characterized Aeromonas strains, is of importance. Representatives of this genus are emerging pathogens and there is a gap in our knowledge of the evolutionary history and taxonomic organization of this genus. The experiments and analysis appear to have been carried out carefully and adequately. That being said, this paper should never have been submitted to any journal in its current form. The abuse of English in this submission is just appalling, to the extent that is it impossible to understand much of what the authors have written. Throughout the manuscript, verbs and whole phrases are missing at critical points, so that understanding what the authors are trying to convey is an exercise in futility. The first two pages of my copy are marked with question marks and notations such as 'something missing?' (or more profanely) 'wtf?'. I gave up marking after that point. The authors need to find a copy editor, or someone who is familiar with scientific writing in English and have them re-write this paper. I'm a native English speaker and have been reading/writing scientific articles for over 40 years; if I can't decipher what the authors are trying to say, non-native speakers will have no hope of reading, much less understanding this MS.

Author Response

The manuscript has now undergone extensive revision throughout the entire document.

Reviewer 2 Report

I would like to thank the authors for providing important information about Aeromonas rivipollensis genomes in order to determine the virulence and antibiotics resistance genes and to define its genetic population structure.

The manuscript is well written, well structured and adequately cited.

The present paper is recommended for publication, after revising its structure. The current structure of the paper “2. Results, 3. Discussion, 4. Materials and Methods and 5. Conclusions” is not helpful for the reader.

Please revise the structure to “2. Materials and Methods, 3. Results, 4. Discussion and 5. Conclusions”

Author Response

The organization has been changed, and section 2 now contains the materials and methods.

Reviewer 3 Report

This study sought to characterize Aeromonas rivipollensis strains isolated from river water in Johannesburg, South Africa. This study seems to demonstrate that A. rivipollensis strains are potentially virulent, an emerging pathogen that the river water can transmit.

Despite this critical observation, the authors must address some areas of concern.

Areas of concern:

General:

English changes are required in different parts of the manuscript.

Introduction

Lines 33-35: this statement should be reformulated to avoid plagiarism

Results

Table 1: should include genome features of the sequenced of another well-known Aeromonas species for comparison purposes

Materials and methods

Line 328: incubated at instead of incubated for.

Author Response

English changes are required in different parts of the manuscript. The English language and style have been revised throughout the entire manuscript

Introduction

Lines 33-35: this statement should be reformulated to avoid plagiarism. This was revised to: They are emerging, opportunistic pathogens that frequently transmit from the environment to humans, causing a wide spectrum of infections.

Results

Table 1: should include genome features of the sequenced of another well-known Aeromonas species for comparison purposes. To facilitate comparison, the Aeromonas rivipollensis KN-Mc-11N1 reference strain was included in Table 1. This is a complete genome of A. rivipollenisis currently available on NCBI.

Materials and methods

Line 328: incubated at instead of incubated for. The amendments were made.

Reviewer 4 Report

The manuscript entitled Comparative genomics revealed a potential threat of Aeromonas rivipollensis G87 strain and its antibiotic resistance” by Fono-Tamo et al. performed whole genome sequencing study to profile the virulence factors and antibiotic resistance genes.  The design and methodology adopted for this study is reasonable, however it seems there is several writing mistakes in the text (for e.g, (i) Line 83-84 the mentioned protein coding sequences for A. rivipollensis genomes are not the same as mentioned in the table 1, (ii) the citation of Table 1 in the text is missing) that require careful revision of this manuscript before publication.

Author Response

Line 83-84 the mentioned protein coding sequences for A. rivipollensis genomes are not the same as mentioned in the table 1. The wording was changed to match the protein-coding sequences in the table that were annotated by the NCBI.

The citation of Table 1 in the text is missing that require careful revision of this manuscript before publication. The results have been improved for clearer representation, and the citation has now been inserted into the text.

Round 2

Reviewer 1 Report

The manuscript has been revised extensively and while not outstanding, the revised version addresses a majority of the deficiencies noted in the original submission.

A few items that still need attention are noted below.

In several instances, the authors are mixing up SA and SA synthesis/catabolism. For instance:

Lines 359-360: 'This was discovered in the annotation subsystem, and it was confirmed that it was an exogenous sialic acid based on the genetic variant code 2.0 assigned to it.'

The sentence should be re-written to correct this.

Lines 229-230 should be re-written as (something like) : The accessory binary genes (Figure 2B) found in A. rivipollensis strain G87 were biosynthetic genes for polysialic acid, which is responsible for capsulation and is also found in KN- 230 Mc-11N1 strain.

Also, line 378:'Organisms that can catabolize  but cannot synthesis in the operon are classified as exogenous sialic acids.' should be re-written as (something like) 'Organisms that can catabolize, but not synthesize sialic acids are classified as exogenous sialic acid scavengers.'

Lines 396-397; insert 'resistance'  'In addition, the genome of isolate G87 was only one of the sequenced A. rivipollensis in this study that showed a multidrug tetracycline resistance gene.'

Lines 420-422: It's unclear what 'accuracy' is referring to; accurate identification ???

Author Response

In several instances, the authors are mixing up SA and SA synthesis/catabolism. For instance:

Lines 359-360: 'This was discovered in the annotation subsystem, and it was confirmed that it was an exogenous sialic acid based on the genetic variant code 2.0 assigned to it.' The sentence should be re-written to correct this.

The sentence was revised in lines 377-380 as " 

Sialic acid is biosynthesized, activated, and polymerized by proteins NeuABCD [49], which are present in G87 genome with the exception of the neuD gene. The neuD is a gene found in organisms that can synthesize sialic acid [50]".

Lines 229-230 should be re-written as (something like) : The accessory binary genes (Figure 2B) found in A. rivipollensis strain G87 were biosynthetic genes for polysialic acid, which is responsible for capsulation and is also found in KN- 230 Mc-11N1 strain.

The sentence was revised as " The accessory binary genes (Figure 2B) found in A. rivipollensis strain G87 were biosynthetic genes for polysialic acid, which is responsible for capsulation and is also found in the KN-Mc-11N1 strain."

Also, line 378:'Organisms that can catabolize  but cannot synthesis in the operon are classified as exogenous sialic acids.' should be re-written as (something like) 'Organisms that can catabolize, but not synthesize sialic acids are classified as exogenous sialic acid scavengers.'

The sentence was corrected to " Therefore, organisms that can catabolize, but not synthesis salic acids are classified as exogenous sialic acids. "

Lines 396-397; insert 'resistance'  'In addition, the genome of isolate G87 was only one of the sequenced A. rivipollensis in this study that showed a multidrug tetracycline resistance gene.'

resistance was added in line 415

Lines 420-422: It's unclear what 'accuracy' is referring to; accurate identification ???

The sentence was revised as " These genomes will be used as a resource for additional research that may reveal new information on the genes responsible for accuracy identification"